# Growth and Distribution of Maize Roots in Response to Nitrogen Accumulation in Soil Profiles after Long-Term Fertilization Management on a Calcareous Soil

**Yunlong Zhang [1,2], Tengteng Li [1], Shuikuan Bei [1], Junling Zhang [1] and Xiaolin Li [1,*]**

[1] College of Resources and Environmental Sciences, China Agricultural University, Beijing 100193, China; zhangyunlong@caas.cn (Y.Z.); bs20173030283@cau.edu.cn (T.L.); beisk@cau.edu.cn (S.B.); junlingz@cau.edu.cn (J.Z.)

[2] National Hulunber Grassland Ecosystem Observation and Research Station, Institute of Agricultural Resources and Regional Planning, Chinese Academy of Agricultural Sciences, Beijing 100081, China

[*] Correspondence: lixl@cau.edu.cn; Tel.: +86-10-62732574

**Abstract:** The replacement of inorganic fertilizer nitrogen by manure is highlighted to have great potential to maintain crop yield while delivering multiple functions, including the improvement of soil quality. However, information on the dynamics of root distributions in response to chemical fertilizers and manure along the soil profile is still lacking. The aim of this study was to investigate the temporal-spatial root distributions of summer maize (*Zea mays* L.) from 2013 to 2015 under four treatments (unfertilized control (CK), inorganic fertilizer (NPK), manure + 70% NPK (NPKM), and NPKM + straw (NPKMS)). Root efficiency for shoot N accumulation was increased by 89% in the NPKM treatment compared with the NPK treatment at V12 (the emergence of the twelfth leaf) of 2014. Root growth at 40–60 cm was consistently stimulated after manure and/or straw additions, especially at V12 and R3 (the milk stage) across three years. Root length density (RLD) in the diameter <0.2 mm at 0–20 cm was significantly positively correlated with soil water content and negatively with soil mineral N contents in 2015. The RLD in the diameter >0.4 mm at 20–60 cm, and RLD <0.2 mm, was positively correlated with shoot N uptake in 2015. The root length density was insensitive in response to fertilization treatments, but the variations in RLD along the soil profile in response to fertilization implies that there is a great potential to manipulate N supply levels and rooting depths to increase nutrient use efficiency. The importance of incorporating a manure application together with straw to increase soil fertility in the North China Plain (NCP) needs further studies.

**Keywords:** root length density; fertilization; compost; straw

## 1. Introduction

Crop yields and quality depend upon substantial nitrogen (N) inputs [1]. Modern agriculture depends on high external inputs of chemical N fertilizers to achieve high crop productivity, whereby excessive N leads to low nutrient use efficiency and is now recognized to be a major cause of environmental problems, including N leaching and soil acidification [2,3]. In intensive cropping systems, the overuse of chemical fertilizers in excess of crop nutrient demand does not increase grain yield. Mining the biological potential of crop roots has been proposed to be one of the possible solutions to increase N efficiency [4,5], as roots are closely associated with soil resource acquisition and are central for achieving a high crop yield. Previous studies have demonstrated that root distribution, architecture, and morphological and physiological traits are important in N acquisition [6]. For example, N use

efficiency was increased by 27–38% as a result of increased maize root length [7]. In addition, roots play important roles in the anchorage of plants, the synthesis of growth regulators, and storage.

Nitrogen in the soil is distributed heterogeneously. Plant roots exhibit large plasticity to the changes in nutrient availability by modifying root growth [8] or root physiological traits [9]. Root length, including that of primary roots, seminal roots, and nodal roots is increased under N-deficient conditions, which assists plant roots to explore a larger soil volume and, thus, increases the spatial N availability [10]. By contrast, both lateral root density and root length are reduced, because insufficient N supply inhibits the growth of axial root elongation [11].

Similarly, previous studies have found that low [12] or high nitrate concentrations [13] reduced lateral root branching density and root elongation. The inhibitory effect of high nitrate concentrations on root elongation was associated with low auxin/nitric oxide (NO) and high cytokinin levels in the roots [14]. Apart from chemical N fertilizers, manure is also an important N source and the mineralization of organic N to inorganic N is mediated by soil microbes. The addition of manure was shown to prevent the leaching of applied chemical N fertilizers [15], but also stimulated root growth due to the rich contents of amino acids and some physiological active substances in the organic fertilizers [16]. Over three consecutive years in a field trial, maize root length density down to a 30 cm depth had the highest value in the NPK + FYM (farm yard manure) treatment, which were 70.5% and 31.9% higher than those in the control and NPK plots, respectively [17].

The North China Plain (NCP) plays an indispensable role in China's food provision, accounting for 32% of national grain production in 2015. Soils in the NCP are historically known to be low in fertility [18], which is a major challenge for crop production in this region. The low SOC (soil organic carbon) content is mainly due to the burning of crop straw after harvest and the insufficient application of organic manure. On the other hand, the excessive use of chemical fertilizers leads to low N and P use efficiency, whilst high accumulations of ions in the soil are detrimental to soil aggregations. With the awareness of environmental problems and the pursuit of sustainable agricultural development, the application of organic manure and/or crop straw is becoming an appealing agricultural practice in China [6], as has also occured in many other countries [19,20]. Organic substrates applied alone or in combination with chemical fertilizers were shown to increase crop productivity [21], SOC content [22], soil aggregation [23], and microbial activities. Unlike chemical fertilizers, nutrients contained in the organic manures are released more slowly and can be absorbed by plants only after being mineralized. Little is known about the temporal and spatial dynamics of root growth along the soil profiles. In the present study, we aimed to investigate root growth in response to different fertilization treatments, with an emphasis on temporal and spatial differences in root distributions between the compost and chemical fertilization treatments. We hypothesized that the amendment of compost-stimulated root growth and altered root distributions along the soil profiles, and deep rooting of maize roots in the organic treatments, might increase the acquisition of nutrients and reduce $NO_3{}^-$-N leaching.

## 2. Materials and Methods

### 2.1. Study Site

The experiment site was located at the Quzhou Experimental Station of the China Agricultural University (36°42′ N, 114°54′ E), Quzhou County, Hebei Province. Quzhou County is a typical area of intensive agriculture on the NCP, and a winter wheat–summer maize rotation is the common cultivation practice. The annual mean precipitation and annual temperature is 494 mm and 13.2 °C, respectively. The precipitation during the maize growing season was 372 mm, 274 mm, and 170 mm in 2013, 2014, and 2015, respectively. The soil type at the study site is a silt loam, with a pH of 7.24 ($H_2O$), total N of 0.90 g $kg^{-1}$, soil organic matter of 13.7 g $kg^{-1}$, available P (Olsen-P) of 12.01 mg $kg^{-1}$, and exchangeable K ($NH_4OAc$-K) of 176.2 mg $kg^{-1}$.



## 2.2. Fertilization and Treatments

The field experiment was established in October 2010, and data were collected over three years: 2013, 2014, and 2015. There were four treatments in four replicates with a plot size of $5 \times 10$ m: (1) an unfertilized control (CK), with no fertilizer input; (2) inorganic fertilizer (NPK), a fertilizer only based on conventional farming practice, with 250 kg N ha$^{-1}$ (urea), 45 kg $P_2O_5$ ha$^{-1}$ (super-phosphate), and 45 kg $K_2O$ ha$^{-1}$ (potassium sulfate), respectively; (3) 70% NPK + 30% compost (3000 kg ha$^{-1}$, NPKM). The NPK fertilizer was replaced by a commercial compost (30% replacement, based on the amount of N fertilizer). The compost was mainly derived from cow wastes with 33.2% carbon, 2.0% total N, 0.8% total P, and 0.7% total K; and (4) 70% NPK + 30% compost + wheat straw (NPKMS). The wheat straw was chopped to approximately 2–4 cm, and then returned to the soil at rates of 6, 5, and 8 Mg ha$^{-1}$ in 2013, 2014, and 2015, respectively, as a result of yield variations among different years.

## 2.3. Sampling and Measurement

The summer maize (cv. 'Zheng dan 958') was sown in mid-June, and the density was controlled at 63,000 seeds ha$^{-1}$. In 2013, 2014, and 2015, three neighboring plants were selected and cut at the base of the stem at V6 (the emergence of the sixth leaf), V12, R3 (grain filling), and R6 (physiological maturity) stages, respectively. The entire shoots of maize plants were dried at 60 °C for 48 h until weighted. Subsamples were ground and digested using $H_2SO_4$-$H_2O_2$. The N content of plant samples was analyzed using the Kjeldahl method. The maize grain yield, the number of grains per spike, and the thousand-kernel weight (TKW) were determined [24].

An auger sampling method [25] was used to assess the temporal and spatial distribution of maize roots over the whole growth period. Two intact soil cores per plot, with a volume of 8 cm in diameter and 60 cm in depth, were collected: one within the rows and one between the rows. They were separated into six subsamples at 10-cm depth intervals (0–10, 10–20, 20–30, 30–40, 40–50, and 50–60 cm). After picking out all of the visible roots in the soil cores, the roots were transferred to the laboratory, washed free of soil, and then frozen at −20 °C for a root length analysis [26]. The roots were placed in a glass rectangular dish (200 mm × 150 mm) for scanning with a scanner (Epson 1680, Indonesia). The root length and root diameter were analyzed using the software WinRHIZO (version 5.0, Canada). After calculation, the root length with different diameters (>0.4 mm, 0.2–0.4 mm, and <0.2 mm) and the total root length in each soil layer were obtained.

In 2015, at the R6 stage after maize roots were picked out, the soil was collected from each soil layer at 10-cm intervals. The soil was sieved with a 3-mm sieve. Soil organic matter (SOM) was determined using the dichromate oxidation method. The available P (AP) was analyzed colorimetrically after the soil was extracted with 0.5 mol dm$^{-3}$ $NaHCO_3$ [27]. The soil was extracted with 1 mol dm$^{-3}$ $NH_4OAC$ (available K, AK), and the K concentration was analyzed by a flame photometer.

In 2015, over the maize growth period, soils were collected from the 0–60 cm soil layers (at 20-cm intervals) in the inter-row area for the measurement of soil $N_{min}$ and water content. The soil samples were sieved with a 3-mm sieve, and extracted immediately with 0.01 mol dm$^{-3}$ $CaCl_2$ solution after transfer to the laboratory. Soil $N_{min}$ ($NH_4^+$-N + $NO_3^-$-N) content was analyzed by a continuous flow analysis (TRAACS 2000, Bran and Luebbe, Norderstedt, Germany). Soil water content was measured by oven drying at 105 °C for 12 h.

## 2.4. Calculation

The root length density (RLD) was defined as the ratio of root length to excavated soil volume. The RLD at soil depths of 0–60 cm over the maize growth period was calculated consecutively for three years. In 2015, in addition to the total RLD, the fractions of RLD in different diameter classes (>0.4 mm, 0.2–0.4 mm, and <0.2 mm) were also analyzed. Root efficiency for shoot N accumulation (REN) was defined as the ratio of shoot N accumulation to total root length [28].

$$\text{RLD (cm cm}^{-3}) = \text{Root length/excavated soil volume} \tag{1}$$

$$\text{REN (mg m}^{-1}) = \text{Shoot N accumulation/total root length} \tag{2}$$

In 2015, after the maize harvest, N surplus and N balance were calculated.

$$\text{N surplus} = \text{NF} - \text{NG (kg ha}^{-1}) \tag{3}$$

where NF and NG represent the amount of fertilizer N and the grain N, respectively.

$$\text{N balance} = (\text{NF} + \text{NR IN}) - (\text{NG} + \text{NR OUT}) \text{ (kg ha}^{-1}) \tag{4}$$

where NF and NG represent the amount of fertilizer N and the grain N, respectively. NR IN and NR OUT represent the N contents in the residues from the previous crop and in the harvested crop, respectively.

### 2.5. Statistical Analysis

Effects of fertilization treatments on maize yield, yield components, shoot biomass, shoot N uptake, total RLD, the RLD in the three diameter classes, soil water content, soil $N_{min}$, SOM, AP, and AK were analyzed using one-way analyses of variance (ANOVA) using the SPSS 18.0 for Windows software package. All the data were tested and log-transformed to achieve normality of distribution and homogeneity of variance before conducting ANOVA. Means of different treatments were conducted by the least significant difference (LSD) test, and all of the means were compared at the 0.05 probability level. The relationships between root length density, soil water content, soil $N_{min}$, and shoot N uptake were conducted using Pearson's correlation through the nonlinear response model.

## 3. Results

### 3.1. Grain Yield, Dry Matter Accumulation, and N Uptake in Maize Shoots

The grain yield in the CK was significantly lower compared to the fertilized treatments over three years (except NPK in 2013). The maize yields in the two compost addition treatments (NPKM and NPKMS) were equivalent to or significantly higher than that in the NPK treatment (Table 1). No significant difference in spike number was observed among the treatments. Compared with CK, fertilization treatments significantly increased the grain per spike (except in 2013, and in the NPK and NPKM treatments in 2014), and no significant difference was observed among the fertilized treatments. The TKW in the CK was significantly lower than that in the NPK (except 2013). The TKW in the compost was lower than that in the NPK treatment, but no significant difference was observed between the NPKM and NPKMS treatments (Table 1).

**Table 1.** Maize yield and yield components under different treatments in the three consecutive years.

| Year | Treatments | Grain Yield (Mg ha$^{-1}$) | Yield Components | | |
|------|-----------|---------------------------|------------------|------------------|-------------------------|
| | | | Spikes ha$^{-1}$ | Grains spike$^{-1}$ | TKW (g per 1000 grains) |
| 2013 | CK | 5.72 ± 0.46b | 51,708 ± 846a | 433.8 ± 25.1a | 282.7 ± 5.7a |
| | NPK | 6.44 ± 0.25ab | 50,433 ± 327a | 452.2 ± 14.7a | 284.5 ± 11.7a |
| | NPKM | 7.10 ± 0.61a | 51,567 ± 732a | 443.3 ± 6.3a | 275.9 ± 8.8a |
| | NPKMS | 6.89 ± 0.27a | 52,471 ± 366a | 449.2 ± 10.6a | 280.2 ± 8.0a |
| 2014 | CK | 5.73 ± 0.16c | 67,292 ± 1296a | 325.3 ± 32.6b | 276.0 ± 6.5b |
| | NPK | 8.54 ± 0.30b | 65,450 ± 818a | 449.4 ± 39.1ab | 314.2 ± 5.8a |
| | NPKM | 9.25 ± 0.57ab | 64,317 ± 1097a | 444.5 ± 54.6ab | 295.4 ± 13.7ab |
| | NPKMS | 9.80 ± 0.38a | 65,167 ± 1388a | 541.8 ± 36.7a | 288.9 ± 6.9ab |
| 2015 | CK | 4.39 ± 0.27b | 50,000 ± 1416a | 207.8 ± 39.3b | 255.1 ± 2.5b |
| | NPK | 7.67 ± 0.34a | 51,806 ± 731a | 381.3 ± 44.1a | 276.2 ± 12.8a |
| | NPKM | 7.63 ± 0.37a | 48,750 ± 765a | 398.9 ± 9.1a | 260.0 ± 11.2ab |
| | NPKMS | 8.35 ± 0.32a | 49,722 ± 1332a | 417.3 ± 47.3a | 261.6 ± 9.3ab |

Mean ± standard error (SE) (*n* = 4). The different letters within a column in each year denote significant differences at *p* < 0.05. TKW, thousand-kernel weight; CK, unfertilized control; NPK, inorganic fertilizer; NPKM, manure + 70% NPK; NPKMS, NPKM + straw.

Regardless of the fertilization treatments, the shoot biomass and N accumulation of maize during the growth period showed similar patterns over all three years. The average values were much higher in the fertilization treatments than that in the CK. No significant difference in biomass accumulation or N uptake was observed among the fertilized treatments (Table 2).

In all three years, the root efficiency for shoot N accumulation (REN) in the fertilized treatments was equivalent to or significantly higher than that in the CK during the maize growth period (Table 3). In 2013, no significant differences among all the treatments were observed, except that REN was higher in fertilization treatments than in the CK at the R3 stage. Compared to the CK, REN in the fertilization treatments was significantly increased at V12 in 2014 and R3 in 2015.

### 3.2. Distribution of Total Root Length Density

After the V6 stage, the total RLD began to increase dramatically and peaked at R3 and then decreased rapidly at the R6 stage. There was no significant difference among different treatments except RLD at V12 in 2014 (Table 3).

The distribution of RLD in different soil depths was consistent at all maize growth stages in all three years, regardless of the treatments (Figure 1). There were many more roots on the surface (0–20 cm) relative to the deeper soil layers, especially at the reproductive stages (V12 and R3). In 2013, no significant difference among fertilization treatments was observed at all soil depths over the growth period, except that at V12 RLD in the NPKM treatment was 144% higher relative to that in the NPK at 50–60 cm. Fertilization had no effect on RLD in each soil depth over the growth period, except that RLD in the NPK treatment at 10–20 cm and NPKMS at 40–50 cm was significantly increased relative to other treatments at V12 in 2014. Fertilization showed no effect on RLD in each soil depth at the V6 stage in 2015. At the V12, R3, and R6 stages, the effect of fertilization on RLD was shown at 50–60 cm. At stage V12, compared to the CK, the addition of compost (NPKM and NPKMS) significantly increased RLD, and RLD in the NPKM was increased by 144%. The RLD in the treatments of NPKMS (at R3) and NPKM (at R6) was significantly higher than that in the CK.

### 3.3. Root Distributions in Diameter Classes

In 2015, we measured the distribution of roots in different diameter classes. Fertilization had a relatively small effect on RLD in the diameter (D) classes of D < 0.2 mm, 0.2 < D < 0.4 mm, and D > 0.4 mm (Tables 4–6). The temporal changes had a relatively small effect on RLD of each diameter in a similar manner. The RLD of each diameter was higher in the upper soil layers (0–20 cm) than that in deeper layers. The RLD in D < 0.2 mm was higher than those of the other diameters, accounting for approximately 50% of total root length. No significant difference among treatments was observed at all soil depths in the three diameter classes at the V6 stage. The effect of fertilization was pronounced at 40–60 cm, where a higher value in the NPKM and NPKMS was observed (Table 4).

For the RLD in 0.2 < D < 0.4 mm, fertilization showed a relatively small effect on RLD except at V12 and R6. At V12, the RLD in the NPKMS treatment was higher than in the other treatments at 20–30 cm. At R6, NPKMS had the highest RLD, which was significantly higher than the other treatments at 40–50 cm (Table 5). For the RLD in D > 0.4 mm, NPKMS had the highest RLD at 20–30 cm and RLD in the NPKMS treatment was significantly increased relative to in the CK at 40–60 cm (Table 6). At R6, the RLD in the NPKM treatment was higher than in the CK at 40–50 cm, and no significant difference was observed in other soil depths.

**Table 2.** The aboveground biomass and N accumulation over the maize growth period under different fertilization treatments in the three years.

| Year | Treatments | V6 | | V12 | | R3 | | R6 | |
|------|-----------|---------|---------|---------|---------|---------|---------|---------|---------|
| | | Biomass Accumulation (Mg ha$^{-1}$) | N Accumulation (Mg ha$^{-1}$) | Biomass Accumulation (Mg ha$^{-1}$) | N Accumulation (Mg ha$^{-1}$) | Biomass Accumulation (Mg ha$^{-1}$) | N Accumulation (Mg ha$^{-1}$) | Biomass Accumulation (Mg ha$^{-1}$) | N Accumulation (Mg ha$^{-1}$) |
| 2013 | CK | 0.31 ± 0.05a | 6.19 ± 0.96a | 2.30 ± 0.43a | 31.95 ± 7.29b | 7.29 ± 1.21b | 58.97 ± 8.73b | 9.78 ± 0.83b | 78.13 ± 6.84b |
| | NPK | 0.33 ± 0.03a | 7.37 ± 0.45a | 2.99 ± 0.34a | 41.86 ± 4.60ab | 9.76 ± 0.35ab | 111.15 ± 4.11a | 11.87 ± 1.07ab | 117.66 ± 14.76a |
| | NPKM | 0.37 ± 0.04a | 7.48 ± 0.55a | 2.90 ± 0.19a | 44.65 ± 4.68ab | 9.17 ± 0.55ab | 102.13 ± 11.36a | 12.73 ± 0.83a | 118.81 ± 4.84a |
| | NPKMS | 0.40 ± 0.05a | 8.12 ± 1.38a | 2.98 ± 0.21a | 50.93 ± 3.18a | 10.33 ± 1.19a | 118.28 ± 15.59a | 13.37 ± 0.78a | 127.69 ± 9.77a |
| 2014 | CK | 0.15 ± 0.02b | 3.28 ± 0.50b | 2.40 ± 0.32b | 27.71 ± 6.24b | 5.86 ± 0.24b | 54.45 ± 3.57b | 11.43 ± 1.37b | 78.99 ± 10.44b |
| | NPK | 0.27 ± 0.03a | 8.64 ± 0.98a | 3.61 ± 0.13a | 61.84 ± 2.49a | 8.14 ± 0.67a | 104.73 ± 7.16a | 17.53 ± 0.46a | 153.58 ± 13.28a |
| | NPKM | 0.26 ± 0.03a | 7.97 ± 0.59a | 3.21 ± 0.4ab | 59.16 ± 7.15a | 7.63 ± 0.55a | 98.84 ± 9.41a | 15.30 ± 0.65a | 157.07 ± 5.90a |
| | NPKMS | 0.29 ± 0.03a | 9.36 ± 0.64a | 3.74 ± 0.24a | 70.43 ± 5.81a | 8.33 ± 0.48a | 106.10 ± 8.24a | 17.89 ± 1.02a | 166.78 ± 14.25a |
| 2015 | CK | 0.08 ± 0.00b | 1.92 ± 0.09b | 2.20 ± 0.51b | 38.50 ± 8.94b | 4.99 ± 0.24b | 34.29 ± 3.62b | 12.17 ± 0.71b | 71.75 ± 6.54b |
| | NPK | 0.22 ± 0.02a | 6.81 ± 0.95a | 4.05 ± 0.32a | 82.24 ± 6.69a | 7.91 ± 0.65a | 110.16 ± 10.33a | 17.88 ± 0.99a | 156.36 ± 8.73a |
| | NPKM | 0.19 ± 0.02a | 5.69 ± 0.70a | 3.95 ± 0.57a | 78.57 ± 8.57a | 9.02 ± 0.56a | 116.37 ± 7.83a | 17.71 ± 0.82a | 141.67 ± 7.72a |
| | NPKMS | 0.24 ± 0.02a | 6.84 ± 0.49a | 4.21 ± 0.49a | 84.56 ± 7.91a | 8.55 ± 0.27a | 105.37 ± 7.27a | 18.29 ± 0.54a | 149.23 ± 6.01a |

Mean ± SE (*n* = 3). The different letters within a column in each year denote significant differences at *p* < 0.05.

**Table 3.** Effects of different fertilization treatments on root length density (RLD) and root efficiency for shoot N accumulation (REN) at soil depths of 0–60 cm in the three years.

| Year | Treatments | V6 | | V12 | | R3 | | R6 | |
|------|-----------|-----------------|-----------------|-----------------|-----------------|-----------------|-----------------|-----------------|-----------------|
| | | RLD (cm cm$^{-3}$) | REN (mg m$^{-1}$) | RLD (cm cm$^{-3}$) | REN (mg m$^{-1}$) | RLD (cm cm$^{-3}$) | REN (mg m$^{-1}$) | RLD (cm cm$^{-3}$) | REN (mg m$^{-1}$) |
| 2013 | CK | 0.16 ± 0.04a | 0.72 ± 0.11a | 0.26 ± 0.08a | 2.11 ± 0.11a | 0.56 ± 0.13a | 2.37 ± 0.95b | 0.45 ± 0.06a | 3.02 ± 0.67a |
| | NPK | 0.10 ± 0.03a | 1.37 ± 0.33a | 0.24 ± 0.06a | 3.20 ± 0.32a | 0.68 ± 0.15a | 2.99 ± 0.69a | 0.22 ± 0.07a | 12.59 ± 5.37a |
| | NPKM | 0.11 ± 0.01a | 1.18 ± 0.19a | 0.36 ± 0.02a | 1.96 ± 0.32a | 0.61 ± 0.08a | 2.78 ± 0.49a | 0.41 ± 0.09a | 5.07 ± 1.03a |
| | NPKMS | 0.10 ± 0.01a | 1.59 ± 0.42a | 0.32 ± 0.10a | 3.25 ± 0.75a | 0.91 ± 0.13a | 2.29 ± 0.25a | 0.30 ± 0.11a | 9.07 ± 2.71a |
| 2014 | CK | 0.09 ± 0.02a | 0.83 ± 0.31a | 0.26 ± 0.05ab | 1.46 ± 0.24c | 0.44 ± 0.14a | 2.65 ± 1.01a | 0.27 ± 0.03a | 4.45 ± 0.78a |
| | NPK | 0.09 ± 0.02a | 1.93 ± 0.73a | 0.34 ± 0.03a | 3.13 ± 0.16b | 0.44 ± 0.09a | 4.04 ± 0.54a | 0.48 ± 0.08a | 5.47 ± 0.48a |
| | NPKM | 0.12 ± 0.03a | 1.34 ± 0.44a | 0.16 ± 0.03b | 5.92 ± 0.72a | 0.58 ± 0.09a | 2.75 ± 0.56a | 0.45 ± 0.09a | 6.29 ± 1.63a |
| | NPKMS | 0.17 ± 0.04a | 1.03 ± 0.26a | 0.28 ± 0.03ab | 4.11 ± 0.39b | 0.47 ± 0.06a | 3.49 ± 0.55a | 0.49 ± 0.16a | 7.78 ± 3.86a |
| 2015 | CK | 0.19 ± 0.16a | 0.67 ± 0.30a | 0.10 ± 0.01a | 7.84 ± 0.98a | 0.74 ± 0.13a | 0.84 ± 0.15b | 0.41 ± 0.07a | 2.89 ± 0.66a |
| | NPK | 0.26 ± 0.04a | 0.43 ± 0.12a | 0.16 ± 0.01a | 8.89 ± 0.53a | 0.76 ± 0.08a | 2.33 ± 0.38ab | 0.47 ± 0.10a | 6.37 ± 2.06a |
| | NPKM | 0.24 ± 0.08a | 0.44 ± 0.09a | 0.28 ± 0.06a | 4.96 ± 1.37a | 0.71 ± 0.13a | 2.98 ± 0.65a | 0.88 ± 0.14a | 2.74 ± 0.41a |
| | NPKMS | 0.31 ± 0.03a | 0.36 ± 0.02a | 0.35 ± 0.13a | 5.86 ± 2.59a | 0.82 ± 0.19a | 2.67 ± 0.65a | 0.61 ± 0.24a | 6.09 ± 3.03a |

Mean ± SE (*n* = 3). The different letters within a column in each year denote significant differences at *p* < 0.05.

**Table 4.** Effects of different fertilization treatments on root length density (RLD) of diameter <0.2 mm (cm cm$^{-3}$) during different growing periods at soil depths of 0–60 cm in 2015.

| Growth Period | Treatments | 0–10 cm | 10–20 cm | 20–30 cm | 30–40 cm | 40–50 cm | 50–60 cm |
|---|---|---|---|---|---|---|---|
| V6 | CK | 0.05 ± 0.03a | 0.02 ± 0.00a | 0.05 ± 0.03a | 0.03 ± 0.01a | 0.01 ± 0.00a | 0.87 ± 0.84a |
| | NPK | 0.26 ± 0.23a | 0.12 ± 0.09a | 0.17 ± 0.13a | 0.24 ± 0.22a | 0.23 ± 0.13a | 0.25 ± 0.09a |
| | NPKM | 0.13 ± 0.07a | 0.13 ± 0.09a | 0.30 ± 0.28a | 0.13 ± 0.06a | 0.30 ± 0.17a | 0.14 ± 0.05a |
| | NPKMS | 0.10 ± 0.04a | 0.15 ± 0.10a | 0.08 ± 0.06a | 0.28 ± 0.21a | 0.50 ± 0.23a | 0.24 ± 0.12a |
| V12 | CK | 0.14 ± 0.03a | 0.08 ± 0.02a | 0.03 ± 0.02b | 0.01 ± 0.00a | 0.02 ± 0.00a | 0.04 ± 0.00b |
| | NPK | 0.07 ± 0.05a | 0.14 ± 0.01a | 0.06 ± 0.02b | 0.10 ± 0.07a | 0.05 ± 0.01a | 0.07 ± 0.03ab |
| | NPKM | 0.10 ± 0.04a | 0.33 ± 0.15a | 0.13 ± 0.04ab | 0.06 ± 0.01a | 0.14 ± 0.08a | 0.18 ± 0.05a |
| | NPKMS | 0.03 ± 0.02a | 0.20 ± 0.12a | 0.56 ± 0.27a | 0.16 ± 0.05a | 0.08 ± 0.04a | 0.12 ± 0.02ab |
| R3 | CK | 0.80 ± 0.03a | 0.89 ± 0.40a | 0.17 ± 0.04a | 0.13 ± 0.03a | 0.13 ± 0.02a | 0.06 ± 0.02b |
| | NPK | 1.05 ± 0.08a | 0.79 ± 0.57a | 0.09 ± 0.01a | 0.07 ± 0.03a | 0.19 ± 0.04a | 0.18 ± 0.07ab |
| | NPKM | 1.11 ± 0.47a | 0.20 ± 0.04a | 0.24 ± 0.01a | 0.19 ± 0.07a | 0.11 ± 0.05a | 0.06 ± 0.02b |
| | NPKMS | 0.98 ± 0.35a | 0.62 ± 0.29a | 0.09 ± 0.03a | 0.32 ± 0.14a | 0.10 ± 0.05a | 0.27 ± 0.05a |
| R6 | CK | 0.37 ± 0.11a | 0.33 ± 0.06a | 0.11 ± 0.05a | 0.30 ± 0.20a | 0.04 ± 0.01b | 0.06 ± 0.03b |
| | NPK | 0.05 ± 0.04a | 0.76 ± 0.29a | 0.09 ± 0.01a | 0.14 ± 0.04a | 0.11 ± 0.01a | 0.18 ± 0.04ab |
| | NPKM | 0.37 ± 0.25a | 0.58 ± 0.35a | 0.15 ± 0.05a | 0.11 ± 0.02a | 0.14 ± 0.06a | 0.32 ± 0.12a |
| | NPKMS | 0.66 ± 0.55a | 0.53 ± 0.23a | 0.09 ± 0.05a | 0.22 ± 0.10a | 0.14 ± 0.08a | 0.08 ± 0.03b |

Mean ± SE (*n* = 3). The different letters within a column in each year denote significant differences at *p* < 0.05.

**Table 5.** Effects of different fertilization treatments on root length density (RLD) of diameter 0.2–0.4 mm (cm cm$^{-3}$) during different growing periods at soil depths of 0–60 cm (at 10-cm intervals) in 2015.

| Growth Period | Treatments | 0–10 cm | 10–20 cm | 20–30 cm | 30–40 cm | 40–50 cm | 50–60 cm |
|---|---|---|---|---|---|---|---|
| V6 | CK | 0.01 ± 0.01a | 0.01 ± 0.00a | 0.12 ± 0.06a | 0.01 ± 0.00a | 0.01 ± 0.00b | 0.35 ± 0.34a |
| | NPK | 0.02 ± 0.01a | 0.04 ± 0.03a | 0.31 ± 0.15a | 0.04 ± 0.03a | 0.07 ± 0.03ab | 0.09 ± 0.03a |
| | NPKM | 0.05 ± 0.03a | 0.03 ± 0.01a | 0.58 ± 0.05a | 0.06 ± 0.03a | 0.06 ± 0.05ab | 0.07 ± 0.02a |
| | NPKMS | 0.03 ± 0.01a | 0.04 ± 0.02a | 0.36 ± 0.20a | 0.07 ± 0.03a | 0.12 ± 0.04a | 0.08 ± 0.04a |
| V12 | CK | 0.06 ± 0.00a | 0.05 ± 0.03a | 0.02 ± 0.01b | 0.01 ± 0.00b | 0.01 ± 0.00a | 0.03 ± 0.01b |
| | NPK | 0.02 ± 0.01b | 0.08 ± 0.00a | 0.03 ± 0.01b | 0.05 ± 0.03ab | 0.03 ± 0.01a | 0.03 ± 0.01b |
| | NPKM | 0.02 ± 0.01b | 0.15 ± 0.06a | 0.05 ± 0.01b | 0.03 ± 0.01ab | 0.05 ± 0.02a | 0.08 ± 0.02a |
| | NPKMS | 0.02 ± 0.01b | 0.11 ± 0.07a | 0.26 ± 0.13a | 0.07 ± 0.02a | 0.05 ± 0.02a | 0.06 ± 0.01ab |
| R3 | CK | 0.64 ± 0.05a | 0.54 ± 0.25a | 0.10 ± 0.02a | 0.06 ± 0.02a | 0.10 ± 0.02a | 0.04 ± 0.01bc |
| | NPK | 0.70 ± 0.09a | 0.34 ± 0.23a | 0.04 ± 0.01a | 0.03 ± 0.01a | 0.09 ± 0.02a | 0.13 ± 0.04ab |
| | NPKM | 0.66 ± 0.28a | 0.13 ± 0.03a | 0.11 ± 0.05a | 0.09 ± 0.03a | 0.06 ± 0.03a | 0.03 ± 0.01c |
| | NPKMS | 0.69 ± 0.26a | 0.37 ± 0.19a | 0.05 ± 0.00a | 0.15 ± 0.06a | 0.06 ± 0.04a | 0.17 ± 0.04a |
| R6 | CK | 0.23 ± 0.03a | 0.19 ± 0.04a | 0.08 ± 0.04a | 0.17 ± 0.12a | 0.03 ± 0.01b | 0.03 ± 0.01b |
| | NPK | 0.02 ± 0.02a | 0.53 ± 0.19a | 0.03 ± 0.01a | 0.07 ± 0.01a | 0.05 ± 0.01b | 0.09 ± 0.02ab |
| | NPKM | 0.19 ± 0.15a | 0.36 ± 0.24a | 0.06 ± 0.03a | 0.04 ± 0.01a | 0.56 ± 0.26a | 0.16 ± 0.07a |
| | NPKMS | 0.42 ± 0.37a | 0.33 ± 0.14a | 0.04 ± 0.03a | 0.11 ± 0.06a | 0.07 ± 0.05b | 0.04 ± 0.02ab |

Mean ± SE ($n$ = 3). The different letters within a column in each year denote significant differences at $p < 0.05$.

**Table 6.** Effects of different fertilization treatments on root length density (RLD) of diameter >0.4 mm (cm cm$^{-3}$) during different growing periods at soil depths of 0–60 cm in 2015.

| Growth Period | Treatments | 0–10 cm | 10–20 cm | 20–30 cm | 30–40 cm | 40–50 cm | 50–60 cm |
|---|---|---|---|---|---|---|---|
| V6 | CK | 0.02 ± 0.01a | 0.01 ± 0.00a | 0.00 ± 0.00a | 0.00 ± 0.00a | 0.01 ± 0.01a | 0.12 ± 0.11a |
| | NPK | 0.01 ± 0.00a | 0.03 ± 0.02a | 0.01 ± 0.00a | 0.01 ± 0.00a | 0.02 ± 0.01a | 0.03 ± 0.01a |
| | NPKM | 0.03 ± 0.00a | 0.02 ± 0.00a | 0.01 ± 0.01a | 0.02 ± 0.02a | 0.01 ± 0.01a | 0.01 ± 0.00a |
| | NPKMS | 0.02 ± 0.01a | 0.04 ± 0.02a | 0.01 ± 0.00a | 0.01 ± 0.00a | 0.02 ± 0.01a | 0.01 ± 0.00a |
| V12 | CK | 0.08 ± 0.03a | 0.04 ± 0.02b | 0.01 ± 0.00b | 0.00 ± 0.00a | 0.01 ± 0.00b | 0.01 ± 0.00b |
| | NPK | 0.05 ± 0.03a | 0.09 ± 0.02ab | 0.04 ± 0.02b | 0.02 ± 0.01a | 0.02 ± 0.01ab | 0.02 ± 0.01b |
| | NPKM | 0.05 ± 0.02a | 0.15 ± 0.04a | 0.06 ± 0.01b | 0.03 ± 0.00a | 0.03 ± 0.01ab | 0.03 ± 0.00ab |
| | NPKMS | 0.02 ± 0.01a | 0.07 ± 0.03ab | 0.13 ± 0.02a | 0.05 ± 0.01a | 0.04 ± 0.01a | 0.04 ± 0.01a |
| R3 | CK | 0.27 ± 0.08a | 0.18 ± 0.07a | 0.10 ± 0.03a | 0.06 ± 0.01a | 0.08 ± 0.02a | 0.04 ± 0.02b |
| | NPK | 0.52 ± 0.14a | 0.18 ± 0.05a | 0.07 ± 0.01a | 0.05 ± 0.01a | 0.08 ± 0.01a | 0.09 ± 0.03b |
| | NPKM | 0.39 ± 0.11a | 0.12 ± 0.03a | 0.13 ± 0.03a | 0.09 ± 0.02a | 0.04 ± 0.01a | 0.04 ± 0.02b |
| | NPKMS | 0.37 ± 0.08a | 0.20 ± 0.07a | 0.06 ± 0.02a | 0.14 ± 0.03a | 0.06 ± 0.03a | 0.17 ± 0.02a |
| R6 | CK | 0.13 ± 0.08a | 0.20 ± 0.03a | 0.06 ± 0.03a | 0.05 ± 0.02a | 0.02 ± 0.01b | 0.03 ± 0.01a |
| | NPK | 0.04 ± 0.03a | 0.37 ± 0.09a | 0.08 ± 0.00a | 0.06 ± 0.00a | 0.07 ± 0.01ab | 0.09 ± 0.00a |
| | NPKM | 0.21 ± 0.16a | 0.29 ± 0.15a | 0.08 ± 0.01a | 0.07 ± 0.01a | 0.17 ± 0.06a | 0.10 ± 0.04a |
| | NPKMS | 0.29 ± 0.19a | 0.37 ± 0.15a | 0.06 ± 0.03a | 0.09 ± 0.02a | 0.05 ± 0.03ab | 0.06 ± 0.03a |

Mean ± SE (*n* = 3). The different letters within a column in each year denote significant differences at *p* < 0.05.

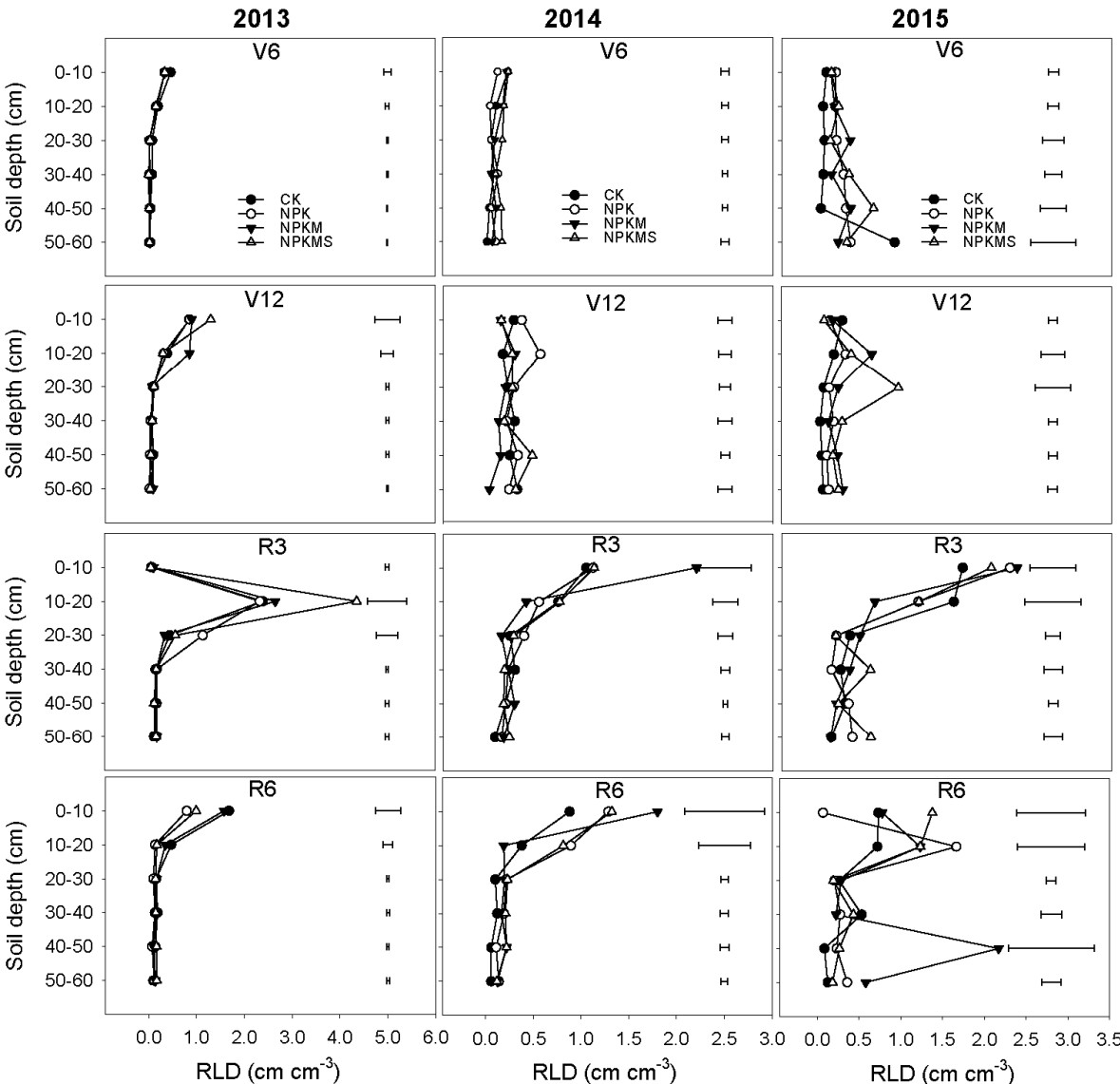

**Figure 1.** Effects of different fertilization treatments on root length density (RLD) during different growing periods at soil depths of 0–60 cm in the three years. The horizontal line is the least significant difference (LSD) ($p < 0.05$).

## 3.4. Soil $N_{min}$, Water Content, and N Balances

In 2015, over the maize growth period, fertilization treatments showed higher soil $N_{min}$ concentrations than CK, and no significant difference was found among fertilization treatments except at V12 and R6 (Figure 2). The average soil $N_{min}$ concentrations in the NPK and NPKMS treatments tended to be higher than those in the NPKM treatment (except at V6). At V12 and R6, at 40–60 cm, soil $N_{min}$ concentrations in the NPK treatment were significantly higher than those in the other treatments.

Soil water contents at 0–20 cm were significantly lower than at 20–60 cm regardless of the fertilization treatments (Figure 2). The soil water contents were significantly affected by the fertilization treatment at the top soil (0–20 cm) but not at deeper soils (except at V12). At the V6 and R6 stages, the soil water contents in the NPKMS treatment were significantly higher than those in the other treatments.

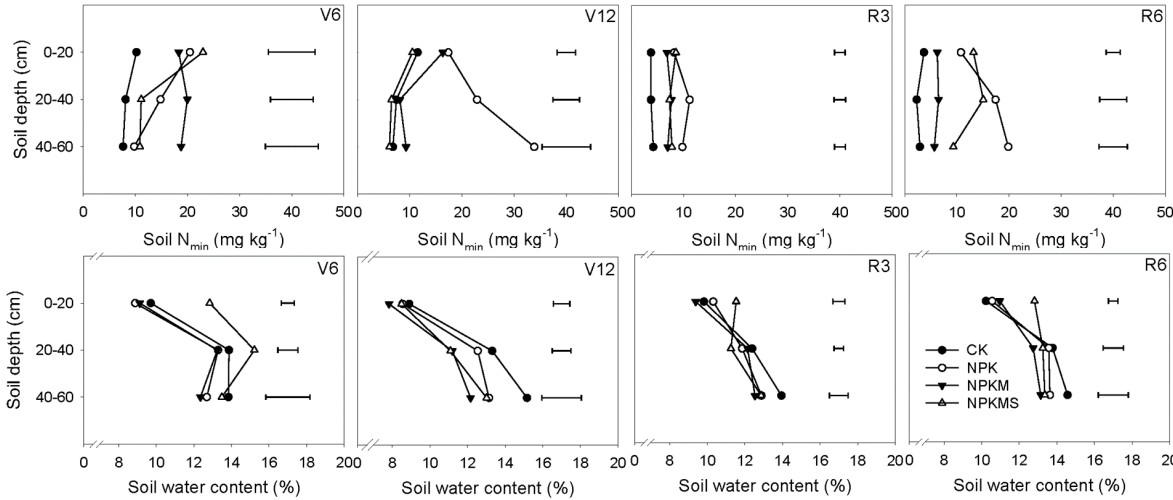

**Figure 2.** Distribution of soil nitrate-N and water contents at soil depths of 0–20 cm, 20–40 cm, and 40–60 cm during different maize growth periods in 2015. The horizontal line is the least significant difference (LSD) ($p < 0.05$).

Both N surplus and N balance were negative in the CK (Figure 3), while the values were positive in the fertilized treatments. The NPK treatment showed the highest N balance (94 kg N ha$^{-1}$) and N surplus (159 kg N ha$^{-1}$), which were 78% and 181% higher than those in the NPKM treatment, and 95% and 57% higher than those in the NPKMS treatment, respectively.

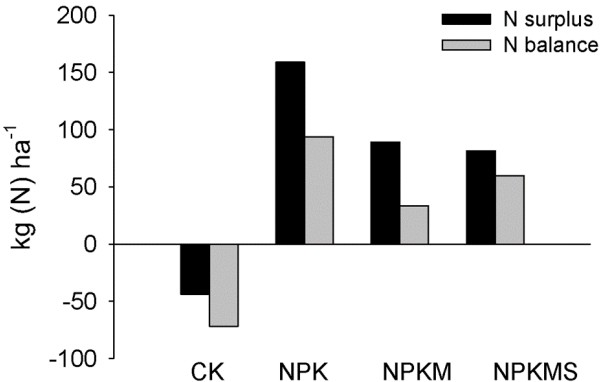

**Figure 3.** N surplus and N balance under different treatments after the maize harvest in 2015.

*3.5. Correlations between Soil Water Content, RLD, Soil N$_{min}$, and Shoot N Uptake*

In 2015, a correlation analysis revealed that soil water contents at 0–20 cm were significantly positively correlated with RLD (D < 0.2 mm) and negatively with soil N$_{min}$ over the maize growth period (Figure 4). RLD (D < 0.2 mm) was significantly positively correlated with shoot N uptake (Figure 4). In deeper soil layers (20–40 and 40–60 cm), root RLD (D > 0.4 mm) showed positive correlations with shoot N uptake (Figure 5a,b).

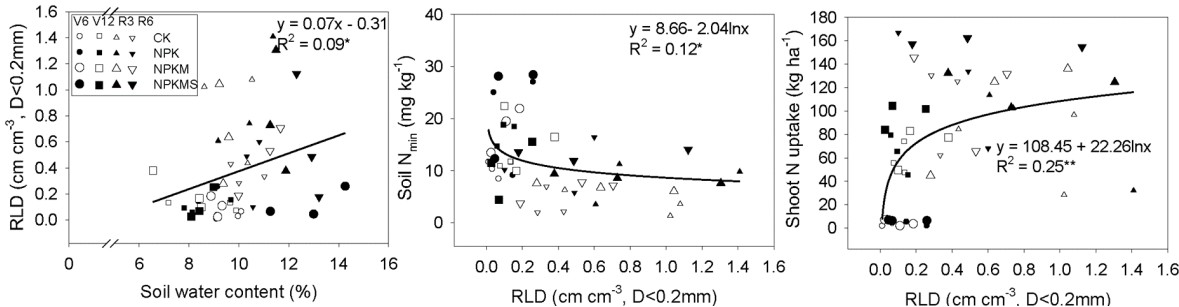

**Figure 4.** Relationships between soil water content, soil $N_{min}$, shoot N uptake, and root length density (<0.2 mm, cm cm$^{-3}$) at 0–20 cm depth in 2015. * $p < 0.05$. ** $p < 0.01$.

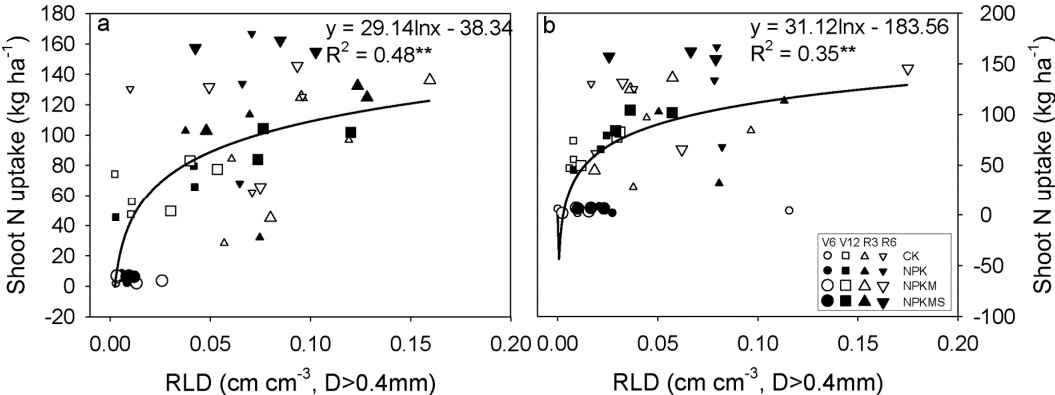

**Figure 5.** Relationships between shoot N uptake and root length density (>0.4 mm, cm cm$^{-3}$) at soil depths of 20–40 cm (**a**) and 40–60 cm (**b**) in 2015. ** $p < 0.01$.

### 3.6. Soil Chemical Properties

The SOM content at 0–20 cm was significantly higher than that at deeper soil layers (Figure 6). At 0–10 cm, the SOM content in the NPKMS treatment was remarkably increased. No significant difference in SOM contents among treatments was observed at 0–10 cm soil. Fertilization had no significant effect on soil AP. The content of soil AP at 0–10 cm and 10–20 cm (8.4 mg kg$^{-1}$) was significantly higher than that in the deeper soils ($p < 0.01$). Available K in the NPKMS treatment was significantly higher than that in the CK and NPK treatments at 0–20 cm, and no significant difference was observed between NPKMS and NPKM treatments. The content of soil AK was significantly higher at 0–20 cm than at deeper soil layers ($p < 0.01$).

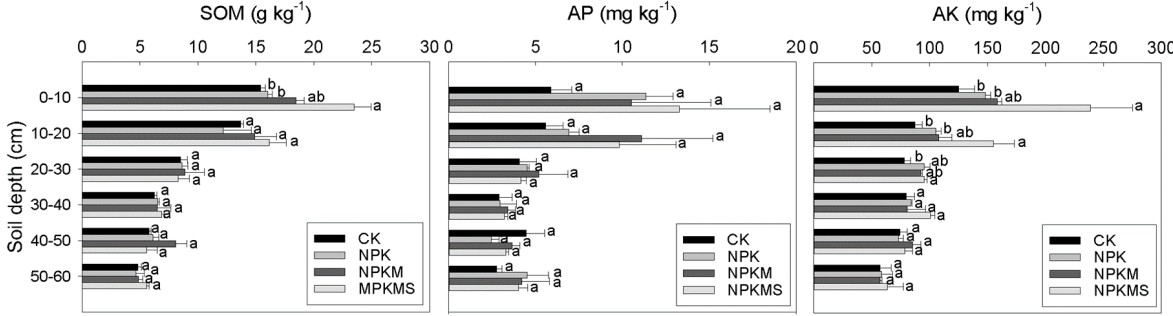

**Figure 6.** Effects of different fertilization treatments on soil organic matter (SOM) and the contents of available phosphorus (AP) and available potassium (AK) at 0–60 cm after the maize harvest in 2015. The different letters within a column in each year denote significant differences at $p < 0.05$.

## 4. Discussion

Nitrate is the most abundant form of available nitrogen in the soil and is acquired by maize plants in great amounts on the NCP. RLD is a significant indicator in estimating nutrient use efficiency, including N [29,30]. The total RLD began to increase dramatically after the V6 stage, reached the maximum RLD at R3, and then declined rapidly at R6 (Table 3). Our results are consistent with other studies relevant to maize root growth [26,31]. Maize plants take up more than half of the nutrients (N, P, and K) during the vegetative growth stage [29–32], and the decrease in the RLD after tasseling indicated rapid root death due to the lateral root mortality [33]. Overall, the fertilization treatments had no significant effect on RLD over all of the seasons, and, also in 2015, in various diameter classes (Tables 4–6). Other than the soil $N_{min}$ content in the whole soil profile, the other soil chemical properties in the top soil layers (SOM and AK) and water contents (Figures 2 and 6) differed greatly among treatments. The correlation between soil $N_{min}$ content and RLD was variable. For example, soil nitrate concentration and maize root length at the silking stage showed a moderate but significant correlation in a 2-year experiment across three types of soils [13]. By contrast, RLD was often weak in acquisition of $NO_3^-$-N [34]. In a long-term field trial, combined N and P fertilizers but not manure applications enhanced nutrient uptake and maize yield via stimulating root growth [35], and the effect was significant at a lower N + P but not at a higher N + P treatment. In the present study, soil $N_{min}$ contents were lower in the manure treatments, yet the value was still relatively high (approx. 16.0–22.0 mg kg$^{-1}$) as shown by the positive N surplus. It is likely that a high N fertilization rate (also high residual N) in the present experiment may have restricted the root growth response. Similarly, high N application may inhibit root elongation at a soil nitrate concentration above 20 mg kg$^{-1}$ [13]. In the present study, we found a weak correlation between RLD and soil $N_{min}$ and soil water content in the upper 20 cm, but a relatively stronger correlation with shoot N uptake at different root classes (Figures 4 and 5), indicating that an N requirement aboveground is the driving force for N acquisition by maize roots. Previous studies found that the amount of N taken up by roots was driven by the demand for shoot growth [33,36]. Root growth was positively correlated with aboveground biomass and N uptake in a solution culture system [10,37]. Our results showed that NPKM and NPKMS treatments could meet the maize N requirement compared with NPK (Table 2). However, the amount of N applied may still be adjusted to allow the roots to mine the soil, and N turnover mediated by soil microorganisms in the organic addition treatments should be taken into account in the organic and compost N management strategies.

Roots were mostly distributed in the upper soil layers over the maize growth stages in all fertilization treatments (Figure 1), and the effect was more significant over time. It is well-acknowledged that the topsoil layer is in favor of root growth, as a result of continual deposition of organic matter and plant residues, fertilization, and cultivation [38]. Maize rooting depth at anthesis varies from approx. 0.7 m to approx. 1 m, and approximately 70% of roots grew in the upper 20 cm of the soil [39]. In general, the effect of fertilization on RLD was more obvious in 2015 after five years of fertilization treatment. The addition of compost increased RLD in certain soil layers. The RLD in the NPKM treatment was 2–4 times higher than that in the NPK treatment at 50–60 cm at the V12 stage, and compost addition increased the RLD relative to the CK at R3 and R6 in 2015 (Figure 1). The higher RLD in different diameter classes was shown in the compost addition treatments (Tables 4–6). The increase in RLD may be associated with the improvement of biological activities due to the rhizosphere effect, which was characterized by exudation of amino acids and some physiological active substances in the organic fertilizer treatment [40]. In addition, the incorporation of organic compost into soil exerts beneficial effects on root growth by improving the physical and chemical properties in the rhizosphere [41,42]. The mechanisms underlying the enhanced growth of plant roots needs further investigations.

In the present study, soil $N_{min}$ content was increased in the NPK treatment over the whole growth period, especially in deep soil layers (20–60 cm), where the highest N surplus and balance was detected (Figure 3). This was in line with the N surplus of >150 kg N ha$^{-1}$ crop$^{-1}$ in a winter wheat–summer

maize rotation experiment conducted in the same region [43]. Our results indicate that NPKM or compost + straw NPKMS met the N requirement for maize growth compared with NPK (Table 2); meanwhile, they maintained crop production and reduced $N_{min}$ leaching. The SOM content in the NPKM treatment at the topsoil (0–20 cm) was increased compared to the CK and NPK treatments (Figure 6). Interestingly, the contents of SOM and AK in the NPKMS treatment were significantly increased compared to the manure-only treatment (Figure 6). Farmyard manure and straw increased SOM content by approx. 37.2% compared to an organic-manure-only or a straw-only application after a long-term (11-year) fertilization in India [44]. Similarly, returning straw and adding farmyard manure significantly increased SOM contents compared with the application of only chemical fertilizer in a long-term (30-year) field experiment in Northwest China [45].

## 5. Conclusions

In summary, this three-year experiment shows that the incorporation of compost to partially replace inorganic N fertilizer (30% in the present study) is feasible to maintain crop yield while reducing the potential N losses. The increase of SOM, available P, and available K concentrations in the compost and straw joint treatment in the top soil highlights the importance of adding fresh plant residues to build up soil fertility, although the underlying mechanism remains to be unraveled. The RLD was not significantly affected by the fertilization treatments, while the vertical distribution of RLD over three years was increased in certain layers in the manure addition treatments, in particular in 2005. In the top soil of 0–20 cm, the RLD was negatively correlated with soil $N_{min}$ contents, and was slightly affected by soil water contents. The RLD at the top soil layer (D < 0.2 mm) and in deeper soils (D > 0.4 mm) was positively correlated with shoot N uptake (Figures 4 and 5). Our results indicate that the partial replacement of NPK by compost might still result in excessive $N_{min}$, and high $N_{min}$ contents in the soil profile may have constrained the response of root growth to fertilization treatments. The N turnover in organic manure and materials mediated by soil microorganisms should be considered in future organic and inorganic N management strategies. The distribution of roots along the soil profile in response to fertilization implies that there is a great potential to manipulate rooting depth to increase nutrient efficiency. The importance of incorporating the application of manure with straw together to increase soil fertility needs further studies.

**Author Contributions:** Y.Z. and J.Z. conceived and designed the experiments; Y.Z., T.L. and S.B. executed the experiments and measured the data; Y.Z. analyzed the data and wrote the paper; Y.Z., J.Z. and X.L. revised the manuscript.

**Funding:** This research was funded by the National Key Research and Development Program of China (2016YFE0101100), the National Basic Research Program of China (2015CB150500) and the Chinese Ministry of Agriculture (201103004)

**Conflicts of Interest:** The authors declare no conflict of interest.

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
