# Peer review of "Growth and Distribution of Maize Roots in Response to Nitrogen Accumulation in Soil Profiles after Long-Term Fertilization Management on a Calcareous Soil"

_sustainability, doi:10.3390/su10114315_

Reviewer 1 Report

The paper aimed to assess the Growth and Distribution of Maize Roots in Response to Nitrogen Accumulation in Soil Profiles after Long-term Fertilization Management on a Calcareous Soil. The topic is of interest and the findings are useful. The experiment has been properly describred. The results have been well reported as well discussed in a good way. I suggest the paper acceptance after an overall revision of English in order to further improve the paper's quality.

Author Response

Thank you for your nice comments. We have thoroughly completed the revision of English in our manuscript.

Reviewer 2 Report

Line 23 correct 20-60 cm

Line 59 correct provision

Line 67 correct [19-20]

Line 93 correct P2O5

Line 94 correct K2O

Line 111 correct 60 °C

Line 122 correct 4-5 mm

Line 128, 130, 133 correct dm-3

Line 136 correct 12 h.

Line  153 correct shoot biomass,

Line 158 correct P < 0.05.

Line  172 correct (Table 1).

Line  204 correct 50-60 cm.

Line  259 correct (0-20 cm)

Line  265 correct (P < 0.05).

Line  268 correct 78%

Line  277 correct (20-40 and 40-60 cm),

Line  281 correct 0-20 cm

Line  284 correct 20-40 cm (a) and 40-60 cm

Line  293 correct 0-20 cm

Line  296 correct 0-60 cm

Line  313 correct NO3

Line  335 correct 1 m,

Line  338 correct 2-4 times

Line  349 correct crop-1

Line  353 correct (0-20 cm)

Author Response

 All the text was revised according to your suggestions.

Reviewer 3 Report

Great job. Please find my comments throughout the manuscript. 

Line 62: 

What does SOC stand for? Soil Organic Compound or Soil organic Carbon? Please add the description.

Line 123: 

Which root morphology characteristics were

recorded? for example: RLD (Root length density) total root length, mean root diameter, and total root surface area

Line 158:

significant differences, and all the means were compared at the 0.05 probability level. 

Author Response

Point 1:

Line 57: What does FYM stand for here? Maybe some readers are not familiar with that so please add the following: FYM (Farm yard manure).

Response: FYM stands for farm yard manure, and it was added into the text.

Point 2:

Line 62: What does SOC stand for? Soil Organic Compound or Soil organic Carbon? Please add the description.

Response: SOC stands for soil organic carbon, and it was added into the text.

Point 3:

Line 123: Which root morphology characteristics were recorded? for example: RLD (Root length density) total root length, mean root diameter, and total root surface area

Response: Here the root morphology characteristics refer to root length and root diameter, and the text was modified.

Point 4:

Line 158: significant differences, and all the means were compared at the 0.05 probability level.

Response: the text was modified according to your suggestions.

Point 5:

Line 30 correct root, fertilization

Line 78 correct Study

Line 88 correct Treatments, Field

Line 107 correct Measurement

Line 152 correct Analysis

Line 153 delete the space

Line 163 correct Yield, Dry, Matter, Accumulation, Uptake, Maize and Shoots

Line 192,193 REN should be consistence using the abbreviation for the same value

Line 195 correct Total, Root

Line 216 correct Diameter

Line 250 correct Water

Line 273 correct Between, Soil, Water, Content, Soil, Shoot, Uptake

Line 285 correct Chemical

Response: the text was modified according to your suggestions.